# Impact and Relevance of the Unfolded Protein Response in HNSCC

**DOI:** 10.3390/ijms20112654

**Published:** 2019-05-30

**Authors:** Olivier Pluquet, Antoine Galmiche

**Affiliations:** 1Institut Pasteur de Lille, Université de Lille, CNRS, UMR8161–M3T–Mechanisms of Tumorigenesis and Targeted Therapies, F-59000 Lille, France; 2Service de Biochimie, Centre de Biologie Humaine (CBH), CHU Sud, 80054 Amiens, France; 3EA7516, Université de Picardie Jules Verne (UPJV), 80054 Amiens, France

**Keywords:** Head and Neck Squamous Cell Carcinoma (HNSCC), Unfolded Protein Response (UPR), therapy, biomarkers, prognosis

## Abstract

Head and neck squamous cell carcinomas (HNSCC) encompass a heterogeneous group of solid tumors that arise from the upper aerodigestive tract. The tumor cells face multiple challenges including an acute demand of protein synthesis often driven by oncogene activation, limited nutrient and oxygen supply and exposure to chemo/radiotherapy, which forces them to develop adaptive mechanisms such as the Unfolded Protein Response (UPR). It is now well documented that the UPR, a homeostatic mechanism, is induced at different stages of cancer progression in response to intrinsic (oncogenic activation) or extrinsic (microenvironment) perturbations. This review will discuss the role of the UPR in HNSCC as well as in the key processes that characterize the physiology of HNSCC. The role of the UPR in the clinical context of HNSCC will also be addressed.

## 1. Introduction

Head and Neck Squamous Cell Carcinomas (HNSCCs) are a group of tumors that arise from the mucosal epithelia of the head and neck. They are often associated with alcohol and tobacco use [1,2] and more recently, for some primary locations, viral etiology (Human Papillomavirus, HPV in oropharynx) [1,2]. Their heterogeneous clinical presentation explains the complexity of the care. Primary surgery is usually the first line for the care. It is limited in its extent by the proximity of vital organs and anatomical structures required for phonation or deglutition. Depending on tumor staging and the presence of histological signs of aggressiveness, it is followed by radiotherapy or radiochemotherapy [3]. Angiolymphatic invasion (ALI) and extracapsular spread (ECS) constitute the main determinant of tumor aggressiveness, together with the presence of invaded surgical margins [3]. Even when the best medical treatments are applied, post-surgical local recurrence occurs in more than half of the cases. It is therefore important to understand how these tumors appear and also how they acquire resistance to their medical treatments.

The molecular analysis of HNSCC has emphasized the remarkable heterogeneity of these tumors [4]. The first landscape analysis of genome alterations, reported in 2015 by The Cancer Genome Atlas (TCGA), found a clear distinction between HNSCC arising in a context of HPV infection and others [5]. While the nature of the alterations identified differ, a common spectrum of driver genes is now identified in HNSCC. The corresponding genes are for example involved in cell cycle control (*CDKN2*, *TP53*, *CCND1*), oncogenic signaling, growth regulation and cell differentiation (*EGFR*, *WNT*), cell survival (*PIK3CA*) and epigenetic regulation [6]. In this respect, HNSCC tumors present common biological features with epidermoid tumors from other primary locations [6].

While genomic analysis provides a unique picture of HNSCC carcinogenesis, it does not reflect the multiple cellular stresses that are encountered by cancer cells within solid tumors. Within the tumor microenvironment, cancer cells experience reduced oxygen and energy supply, oxidative stress, acidosis, which compromise protein folding in the endoplasmic reticulum (ER) and lead to the activation of the UPR [7,8]. This activation has been implicated in various processes of tumor cell biology including angiogenesis, treatment resistance, invasion, aggressiveness and inflammation [9]. Tumor cells often display constitutive UPR activation to favor tumor growth, allowing them to adapt under unfavorable conditions [10]. In the context of HNSCC, chronic UPR activation may therefore represent an interesting aspect for the development of novel therapies.

## 2. Linking HNSCC Carcinogenesis to the UPR

The endoplasmic reticulum, a complex cellular organelle, is a key site for lipid synthesis, calcium storage and for folding, modification of nascent polypeptides encoding transmembrane or secretory proteins. In the ER lumen, nascent proteins interact with chaperones (such as GRP78) to prevent ER exit of incompletely folded proteins. Folding includes signal sequence cleavage, post-translational modifications, such as N-glycosylation and disulfide bond formation through the activity of protein disulfide isomerases (PDI) [11]. Moreover, the high concentration of calcium and oxidizing ER luminal environment facilitate protein assembly [12,13]. Conditions favoring accumulation of misfolded proteins within the ER lumen such as calcium depletion, oxidative stress, microenvironmental stress, perturb ER functions and lead to a state referred to as ER stress [14]. To maintain ER homeostasis, signaling sensors initiate signaling cascades that inhibit protein synthesis, upregulate chaperones and the expression of folding enzymes and induce degradation of misfolded proteins. These adaptive signaling pathways constitute the Unfolded Protein Response (UPR) that involves at least three ER membrane resident proteins: PERK, IRE1α, ATF6α (Figure 1). Accumulating evidence shows that the UPR functions may regulate various processes other than protein folding including proliferation, metabolism and cell death [15]. Chronic ER stress is often found in cancers [10]. Chronic UPR activation has been linked to angiogenesis, tumor aggressiveness, invasion and metastasis [16,17,18]. Cancer cells exploit and activate the ER-resident machinery, leading to aberrant expression of UPR effectors in order to prevent cell death. Nevertheless, severe or prolonged UPR signals can trigger cell death [19]. Thus, the next sections of this review will describe the known UPR protein alterations and their role in HNSCC.

### 2.1. GRP78

GRP78 (glucose-regulated protein, 78kDa) functions as a chaperone for the folding of newly synthesized proteins in the ER but also in the sensing and response of misfolded proteins. GRP78 is a common target of all three arms of UPR upon ER stress induction (Figure 1). GRP78 has been extensively studied and its expression levels are often highly elevated in various cancers [10]. Elevated levels of GRP78 were associated with metastatic potential in numerous cancers [20,21,22]. Overexpressed GRP78 in patient samples are frequently detected in HNSCC [23,24], including nasopharyngeal carcinoma (NPC) [25], oral lesions [26], laryngeal SCC [27], tongue cancer [28] and advanced hypopharyngeal squamous cell carcinomas (HSCC) [24]. Samples with the lowest GRP78 expression were associated with advanced stage of OSCC and neck lymph node metastasis [29]. Although GRP78 is induced in HNSCC, its clinicopathological significance as a prognostic marker remains controversial. For example, GRP78 overexpression positively correlates with a more aggressive potential and poor survival in oral lesions [26] and tongue cancer [28], suggesting that GRP78 may be a prognostic factor in these two cancer types. In contrast, the highest GRP78 expression in HSCC patients or laryngeal SCC patients with advanced disease was found to contribute to better survival [24,27]. The origin of this discrepancy in survival remains to be explored, in particular the impact of GRP78 localization, since this chaperone can be found in the ER lumen, as well as on the cell surface of tumor cells [30]. GRP78 has been extensively studied in cancer. Using data from The Cancer Genome Atlas (TCGA), we addressed the clinical relevance of GRP78 overexpression in HNSCC. Kaplan-Meier analysis/log-rank test for both disease-free survival (DFS) and overall survival (OS) showed that lower GRP78 expression in the cohort has a modest effect on OS (Figure 2). The role of GRP78 in tumor aggressiveness deserves to be further explored in HNSCC.

### 2.2. PERK Pathway

Little is known about aberrant PERK signaling in HNSCC. Analysis of PERK mutations revealed that the gene is rarely mutated in HNSCC (Figure 3). A variability of basal mRNA and protein expression levels of PERK was observed in a panel HNSCC cell lines [31]. However, it is largely recognized that these levels are not indicative of PERK activity. PERK phosphorylates eIF2α, that in turn blocks the global mRNA translation. This results in Cap-independent translation of specific transcription factors such as ATF4 and its target genes ATG5 (encoding a protein related to autophagy) and CHOP (encoding a pro-apoptotic protein). A recent article demonstrated that DDX3, an important component of mRNA metabolism, positively regulates ATF4 expression and a set of its downstream targets, resulting in increased migration and invasion in oral cancer cell lines [32]. Elevated CHOP expression was detected in HNSCC compared to normal oral mucosa [23]. However, several kinases can mediate eIF2α phosphorylation leading to ATF4 activation [33], which could limit the impact of PERK in the observed effects. Nevertheless, PERK silencing in HNSCC cell lines subcutaneously injected in severe combined immunodeficiency (SCID) mice resulted in a significantly decreased tumor growth [23], suggesting a crucial role of PERK in HNSCC tumor growth. PERK also phosphorylates the transcription factor NRF2 [34], which has multiple pro-tumorigenic functions and promotes survival [35]. NRF2 is a key regulator in cellular defense against oxidative stress [36] and its activation is associated with alcohol consumption and tobacco smoking [36], two crucial determinants in the carcinogenesis of HNSCC. The in vivo relevance of NRF2 was demonstrated in an animal model of oral carcinogenesis with 4-Nitroquinoline 1-oxide (4NQO). Indeed, *NRF2*-KO mice were more susceptible to tongue carcinogenesis induced by 4NQO than their wild-type (WT) counterparts [37]. A single mutation of *NRF2* gene was identified in tongue SCC [38] and NRF2 levels are elevated in HNSCC [39] and HNSCC stem cells [40]. NRF2 is also overexpressed in OSCC tumors but was not found to be significantly associated with lymph node metastases and pathological grade [41]. Additionally, a gene signature regulated by the KEAP1-NRF2-CUL3 axis was associated with poor prognosis in HNSCC [42]. However, the impact in NRF2 activation by PERK remains underexplored in HNSCC.

### 2.3. ATF6α Pathway

ATF6α is thought to promote cell survival. We noted that a very low mutation rate of ATF6α occurs in HNSCC (Figure 3). Once activated, ATF6α migrates to the Golgi where it is cleaved by S1P and S2P proteins releasing a cytoplasmic active ATF6α form that shuttles into the nucleus to act as a transcription factor [10]. Numerous genes were transactivated by ATF6α, including GRP94 (glucose-regulated protein, 94 kDa), CRT (calreticulin) and VCP (valosin-containing protein) [43]. GRP94 was found to be overexpressed in oral cavity cancers [44] and nasopharyngeal cancer [45]. Immunohistochemistry (IHC) experiments showed that CRT was differentially up-regulated in OSCC samples [46], maxillary sinus SCC [47] and laryngeal squamous cell carcinoma (SCC) lesions [48]. This elevated CRT expression was also supported by mass spectrometry analyses in OSCC [49]. One study demonstrated that elevated VCP expression (determined by IHC) was associated with better survival and could be a prognostic marker in OSCC [50]. VCP was thought to be an oncogene that drives the increase in DNA copy number at chromosome 9p13, a region associated with oral invasive lesions [51]. Furthermore, activation of ATF6α and PERK have been shown to be essential for the long-term survival of dormant HEp-3 cells (Human Epidermoid carcinoma) derived from a cervical lymph node (metastatic site) from a buccal mucosa squamous cell carcinoma (primary site) in vitro and in vivo [52,53].

### 2.4. IRE1α/XBP1 Pathway

IRE1α is known to promote both cell survival and cell death, its gene is rarely mutated in HNSCC (Figure 3). IRE1α is an ER transmembrane protein with both kinase and endoribonuclease activities. The only known kinase substrate is itself. Historically, XBP1 was described as the first substrate of IRE1α. XBP1 mRNA is processed with the t-RNA ligase RTCB leading to unconventional mRNA splicing [54]. This resulted in a shift in the open reading frame and led to the translation of an active and stable protein named XBP1s, that acts as a transcription factor [55,56]. XBP1 is frequently detected in nasopharyngeal carcinoma (NPC) cancers [25]. When dichotomized, the lowest XBP1 expression is associated with poor prognosis and poor survival in patients with OSCC [57]. However, these IHC analyses did not discriminate between total XBP1 and XBP1s proteins, the latter reflecting XBP1 activity. Nevertheless, a report demonstrated that XBP1 silencing sensitized the OSCC cell line Tca-8113 to cell death by apoptosis, suggesting its crucial role [58]. Derlin-1 (a XBP1s target) is overexpressed in HNSCC and its high expression is positively correlated with lymph node metastasis, clinical stage, disease recurrence and shorter survival [59]. Calnexin, another XBP1s target, is often up-regulated in patients with maxillary sinus SCC [47] and laryngeal squamous cell carcinoma (SCC) lesions [48].

## 3. Linking Specific Aspects of HNSCC Physiology to the UPR

### 3.1. UPR and Angiogenesis

The UPR plays a role in the control of angiogenesis. This aspect has been well reviewed in References [8,10,60,61]. Like other solid tumor types, UPR activation is associated with an up-regulation of proangiogenic factors in HNSCC [23]. The PERK/ATF4 and GCN2/ATF4 pathways have been involved in UPR-mediated angiogenesis in HNSCC cell lines [23,62]. Moreover, the relevance of the GCN2/ATF4 pathway promoting tumor growth and angiogenesis was demonstrated in a xenograft model using HNSCC cells [62]. The impact of UPR in angiogenesis in HNSCC deserves to be further explored.

### 3.2. UPR and Tumour Metabolism

Increasing evidence shows that the UPR plays a major role in the regulation of tumor metabolism. Although the link between the UPR and glucose homeostasis has been little explored in HNSCC, several findings support a regulatory role of the different UPR arms in glycolysis [63,64,65] as well as in the hexosamine pathway (HBP) (reviewed in References [66,67,68]). Recently, two papers described a role of the UPR in amino acid metabolism. Indeed, ATF4 activation resulted in increased amino acid uptake upon glutamine deprivation [69]. Moreover, a low protein diet in tumor-bearing mice resulted in an induction of the IRE1α pathway in cancer cells [70].

### 3.3. HNSCC, UPR and Inflammatory Stroma

HNSCC is among the tumors with the highest levels of stromal immune infiltration [71]. The tumor microenvironment includes various cell types that together can account for a large fraction of most solid tumors. Besides vascular cells, other cell types are present including Cancer-Associated Fibroblasts (CAF) and immune cells, especially Tumor-Associated Macrophages (TAM) and Tumor-Infiltrating Lymphocytes (TIL). These non-malignant cell types support the growth of the cancer cells and often modulate the tumor response to various medical treatments. Activated CAFs typically establish a state of chronic inflammation, sometimes compared to chronic wounding, that overall support tumor growth and invasion/metastasis. Several recent studies point to the UPR in cancer cells as a determinant of the composition and function of the tumor microenvironment in solid tumors.

Studies addressing IRE1α signaling in colorectal cancer and melanoma suggest that the UPR constitutes a robust activation signal toward the immune microenvironment [72]. A direct control of the secretome of tumor cells, typically observed upon induction of the RIDD (Regulated IRE1-Dependent Decay), changes the pattern of cytokines produced by tumor cells. Active mechanisms of innate immunity are also found to be engaged downstream of the UPR. During the RIDD, IRE1α activation produces multiple cleavages of mRNA encoding secreted proteins, potentially leading to RIG-1 (Retinoic Acid-inducible Gene-1) activation. By so doing, the activation of IRE1α mimics a process normally involved in RNA recognition of intracellular pathogens; a process proposed to foster CD8 T cell-mediated anti-tumor immune responses [70]. The role of the UPR is also not limited to malignant cells and it is likely to be important in immune cells, as it plays a fundamental role in the normal development of a number of immune cell lineages, such as B cells or dendritic cells (reviewed in Reference [73]). The perturbation of ER homeostasis in infiltrating immune cells could impede the development of an effective anti-cancer immunity [74]. Surprisingly, recent studies indicate that a transmission of the ER stress is possible between different cells within a tumor. Although evidence for transmissible ER stress (TERS) is lacking for HNSCC at this stage, the findings open up the possibility that a UPR might be transferred between different cells within a solid tumor, possibly leading to a systemic response. Interestingly, the study of the UPR and its regulation of the tumor immune microenvironment illustrates the complex interplay of nutritional, metabolic and environmental determinants. The protein-content of diet is for example reported to modulate the UPR [70].

The study of the protein tumor marker SCCA (Squamous Cell Carcinoma Antigen) provides an interesting case to support the contribution of the UPR to local tumor inflammation in HNSCC. Briefly, SCCA is a tumor product encoded by the *SERPINB3* and *SERPINB4* genes, whose tissue expression and serum concentrations increase in patients with HNSCC [75]. The two genes that encode SCCA are evolutionarily duplicated serine/cysteine protease inhibitors with high amino-acid sequence similarity. They have the shared ability to block cytosolic proteolysis. As a consequence of the inhibition of proteasomal/lysosomal protein degradation, SERPINB3/B4 have been reported to induce the PERK and ATF6α arms of the UPR in a large variety of tumor cells [76,77]. The ability to neutralize intracellular proteases and induce the UPR was suggested to contribute to the installation of a pro-inflammatory phenotype characterized by high expression of interleukin-6 (IL6) [76,77]. This link was specifically addressed in HNSCC by Saidak et al. using transcriptomic data available from TCGA [78]. In HPV-negative HNSCC, a strong association was found between SERPINB3/B4 mRNA levels and the presence of a denser tumor immune infiltrate, an IFN-Gamma signature and the expression of the immune checkpoint regulator PD-L1 [78]. In this situation, the induction of a chronic UPR might be one important determinant of local tumor inflammation, with potential implications for carcinogenesis and the response to treatment.

## 4. UPR and Response to Current HNSCC Treatments

### 4.1. UPR Linked to Radiotherapy

Radiotherapy is often used to treat head and neck cancers, especially for advanced forms. A link between the UPR and radiosensitivity of HNSCC cells was reported by Foy et al. [79]. Importantly, these authors have attempted to identify the genes whose expression in HNSCC might be linked to radioresistance. In HPV-negative HNSCC, a 13 gene-expression score was constructed, that was shown to correlate with the expression of UPR genes [79]. Proteomic studies showed that GRP94 is overexpressed in radioresistant nasopharyngeal (NPC), oral and head and neck cell lines. Silencing of GRP94 in NPC cell lines displayed a reduction in clonogenic survival and restored radiosensitivity [44,80,81]. Xenografted mice with siGRP94-radioresistant cells subjected to radiation treatment showed an increase in radiosensitivity compared to radioresistant parental cells, underlying the critical role of GRP94 in vivo [81]. GRP94 is considered as a poor prognostic indicator in patients receiving radiotherapy [44]. The clinical significance of GRP94 was further demonstrated in patients, where high GRP94 expression positively correlates with resistance to radiotherapy [45]. Other UPR markers seem to play a role in radioresistance. The ATF6-target protein calreticulin accumulated in the nuclei of irradiated SQ-20B cells (HNSCC), although its functional role in radioresistance remains unclear [82]. Additionally, treatment combining 2-Deoxy-D-glucose (2-DG) and 6-aminonicotinamide (6-AN) that decreased NRF2 protein levels led to radiosensitization of a head and neck cell line [83]. Similarly, inhibition of NRF2 by siRNA in the radiation resistant HSC-4 cell line led to decreased cell viability compared to their control counterparts. Subcutaneous injection of HSC-4 cells showed that tumor size, volume and weight were significantly reduced upon radiotherapy and more obvious when NRF2 was silenced [84].

### 4.2. UPR Linked to Chemotherapy

Chemotherapy, especially systemic treatment using the alkylating agent cisplatin, is an essential component of the medical treatment of advanced stages of HNSCC. Few studies have been devoted to the role of the UPR in HNSCC in this context. Actually, cell signaling downstream of ER stress was found to be an important determinant of chemosensitivity/chemoresistance in multiple types of solid tumors other than HNSCC [10,85]. Despite the large amount of studies published to date, only limited supporting evidence exists, because the studies so far have been based on immunohistochemical analysis of one determinant (usually the chaperone GRP78) in small cohorts of cancer patients treated with chemotherapy. Cell line studies on the other hand strongly suggest a general protective role of cancer cells and therefore a role of the UPR in chemoresistance [10]. It is usually accepted that chemotherapeutic agents induce objective responses, that is, tumor regression, by inducing a lethal stress that leads to programmed cell death of cancer cells. In recent years however, this simplistic view of the mode of action of chemotherapy was challenged, especially because of recent research centered on the immunology of cell death. It is today increasingly recognized that part of the efficacy of chemotherapeutic agents comes from other regulated responses, for example relying on the anti-tumor immune response. The UPR might be important in this respect, because triggering ER stress is emerging as a central event for a variety of immunogenic cancer cell death routines [86]. Exposure of the ER chaperone calreticulin on the surface of cancer cells may for example occur as a consequence of ER stress induced by cisplatin [87]. Whether exploring the UPR could help to anticipate the efficacy of chemotherapeutic agents in HNSCC or at least some specific facets of its mode of action, deserves to be investigated in future studies.

### 4.3. UPR Linked to Targeted Therapies/IMMUNE Checkpoint Inhibitors

It has been demonstrated that cetuximab (the only targeted therapy indicated in HNSCC) enhances cisplatin-induced apoptosis by activating ER stress markers such as CHOP in LSCC cells [88]. Another study showed that afatinib stimulated the PERK-eIF2α-ATF4 branch in HNSCC cells leading to the induction of apoptosis [89]. Surprisingly, the inhibition of glycolysis with 2DG enhanced the cytotoxic effect of erlotinib in an HNSCC cell line but, in contrast, suppressed the antitumor efficacy of erlotinib in an HNSCC xenograft mouse [90], probably due to increased autophagic activity mediated by ER stress.

The introduction of immune checkpoint inhibitors, especially targeting the PD1/PD-L1 interaction, constitutes a breakthrough in the medical treatment of recurrent/metastatic HNSCC [91]. Following the results of the two randomized phase 3 trials CHECKMATE-141 and KEYNOTE-040, showing a beneficial effect on OS, nivolumab and pembrolizumab were approved for this indication [91]. The biological rationale for this strategy lies in the relief of the inhibitory interaction between PD-L1, present on the surface of malignant cells and PD1, present on the surface of the lymphocytes that infiltrate the tumor. Interestingly, a recent study opens up the possibility that PD-L1 protein expression at the surface of lung SCC might be correlated with the protein levels of eIF2α and ATF4 [92]. This correlation opens up the interesting prospect that PD-L1 expression on the surface of tumor cells might be linked to the UPR and cell proteostasis in general. Interestingly, this regulation could be targeted by other clinically approved drugs, such as the antidiabetic drug metformin, that was recently reported to target PD-L1 for protein turn-over by endoplasmic reticulum-associated degradation (ERAD) [93].

## 5. The UPR as a Therapeutic Target

### 5.1. Increasing UPR Activity to Promote Cell Death

Because of the highly secretory nature of cancer cells, it has been suggested that cancer cells may have a lower threshold for the activation of ER stress induced cell death. Various chemical or natural products were reported to enhance ER stress and to exert a UPR-dependent cytotoxic effect in HNSCC (summarized in Table 1) [94]. Some HIV protease inhibitors such as lopinavir and ritonavir have been shown to radiosensitize HNSCC cell lines by inducing ER stress/UPR-induced cell death [95].

### 5.2. Inhibiting UPR Markers to Sensitize Cancer Cells to Treatments

The selective inhibition of the UPR arms is also a promising therapeutic target for many cancers, including HNSCC [15]. Several pharmacological inhibitors have been designed to target specific UPR pathways, with the advantage of sensitizing cancer cells to other therapies [7]. Moreover, silencing of VCP (XBP1s and ATF6α target) or calreticulin (ATF6α target) resulted in reduced proliferation rate and decreased anchorage-independent growth in soft agar in oral cancer cells [46,51]. It is worth noting that HNSCC stem cells are sensitive to cell death when GRP78 is targeted [120].

## 6. Conclusions

Taken together, the data presented above indicate that ER stress and UPR signaling play a functional role in HNSCC by regulating key tumor biology processes including progression of the disease and therapy resistance. It has been well established that the UPR can exert positive selection on cancer cells in solid tumors [60,61,121,122,123]. The above findings suggest a therapeutic potential of targeting the UPR machinery in HNSCC: either by increasing misfolded proteins to overwhelm the ER folding capacity, resulting in cancer cell death or by inhibiting the adaptive UPR signals in order to sensitize cancer cells to conventional/targeted therapies.

Considering the importance of a large panel of UPR proteins in HNSCC, one interesting question that arises is to determine whether the UPR status may be used as a prognostic biomarker. Very few in vivo data are available, and the global status of ER stressors and downstream targets are largely unknown. Moreover, multiple parameters that are frequently encountered in cancers including microenvironmental challenges (hypoxia, nutrient or glucose deprivation, changes in pH), cancer type (highly secretory or not), tumor heterogeneity, adjuvant therapy and numerous layers of complexity that may reflect the kinetics and strength of the UPR response. Nevertheless, a recent study demonstrated that in patients with ERα+ breast cancer receiving tamoxifen, UPR overexpression (based on a genomic UPR target genes index) was predictive of reduced survival, cancer recurrence and resistance to tamoxifen therapy [124]. The relationship between the UPR status and clinicopathologiocal features and the prognosis of HNSCC tumors may represent an interesting avenue to better characterize HNSCC carcinogenesis and other types of cancers.

## Figures and Tables

**Figure 1 ijms-20-02654-f001:**
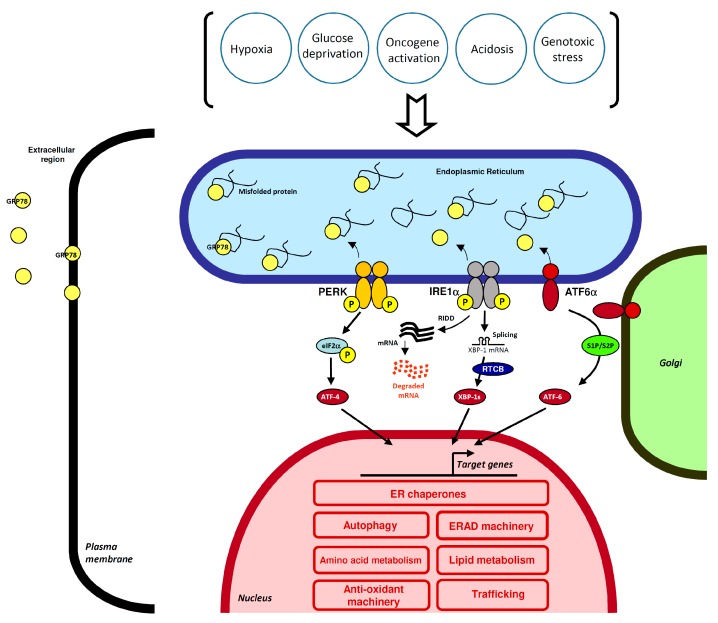
Activation of the UPR pathway in tumor cells. Cancer cells are challenged by intrinsic (oncogene activation) or extrinsic stresses (hypoxia, nutrient and oxygen deprivation, acidosis). These conditions lead to an accumulation of misfolded protein in the ER lumen, a condition referred to as <ER stress>. To overcome such challenges, cancer cells set up adaptive mechanisms including the pro-survival UPR pathway. Three UPR transducers were released from GRP78 to allow their dimerization/oligomerization or export to the Golgi. PERK kinase phosphorylates and activates the transcription factor NRF2 and phosphorylates the translation initiation factor eIF2α, leading to a global suppression of protein synthesis. Paradoxically, this phosphorylation activates the transcription factor ATF4. IRE1α activation, upon oligomerization and trans-autophosphorylation, cleaves the XBP1 mRNA in presence of the t-RNA ligase RTCB to give an unconventional spliced form named XBP1s. XBP1s protein acts as an active transcription factor. IRE1α also promotes JNK activation and controls RNA degradation of various mRNA through its RIDD (Regulated IRE1-Dependent Decay) function. ATF6α translocates to the Golgi, where its cytosolic domain is released upon cleavage by S1P and S2P proteases, to generate an active transcription factor. The three transcription factors coordinate the expression of genes coding for ER chaperones, lipid synthesis, endoplasmic reticulum-associated degradation (ERAD), autophagy machinery, antioxidant response, trafficking, in order to maintain ER homeostasis and control the survival/cell death balance. It is worth noting that GRP78 is also found on the cell surface and is secreted into the circulation favoring interactions with other cells and components of the tumor microenvironment. This opens up new avenues on the role and function of the UPR.

**Figure 2 ijms-20-02654-f002:**
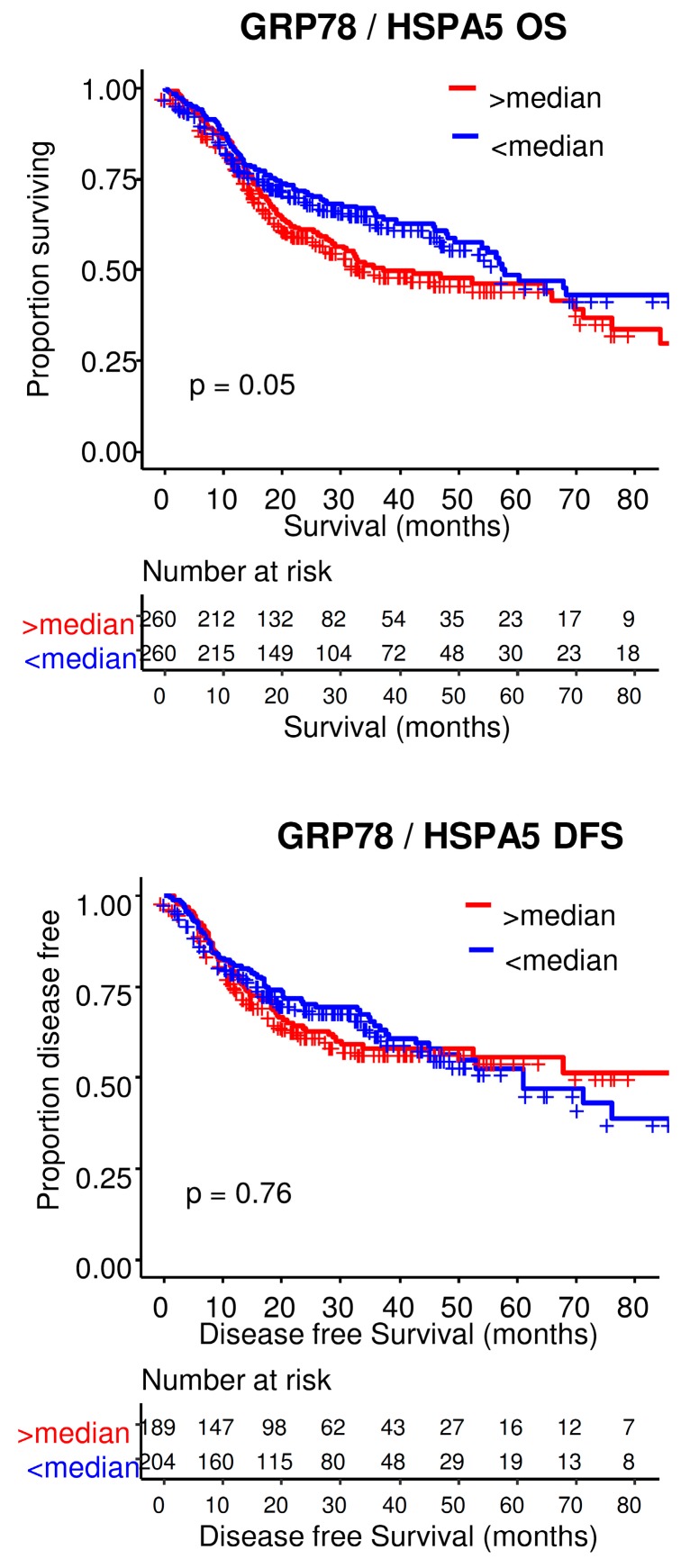
Role GRP78 expression in HNSCC prognosis. Kaplan-Meier analysis comparing the disease-free survival (DFS) and overall survival (OS) in HNSCC tumors for which survival data are available, stratified according to high and low GRP78 expression.

**Figure 3 ijms-20-02654-f003:**
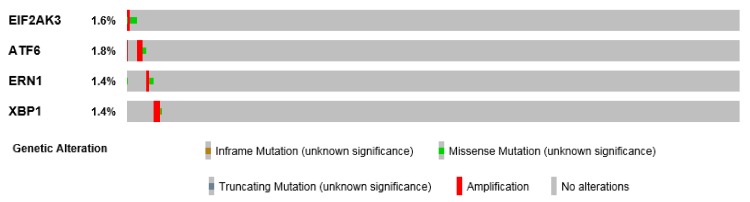
UPR sensor mutation in HNSCC. Spectrum of mutations found in *EIF2AK3* (*PERK*), *ATF6*α, *ERN1*(*IRE1*α) and *XBP1* analyzed from BioPortal and distributed according the type of mutation.

**Table 1 ijms-20-02654-t001:** Natural compounds/drugs showing activation of the UPR-dependent cell death in HNSCC.

Compound (N = Natural/ S = Synthetic)	UPR Effect	Type of HNSCC	Ref
Benzethonium chloride (S)	Increased *CHOP, XBP1* and *ATF3* gene expression; Increased CHOP, ATF4 and GRP78 protein levels	laryngeal human, UMSCC23 cell line	[96]
Borrelidin (N)	Increased XBP1 splicing; Phosphorylation of eIF2α; Increased CHOP, ATF4 mRNA levels	UMSCC-1, -14A and -23 (OSCC) cell lines	[97]
Bortezomib (S)	Increased GADD34, ATF4, protein levels; Phosphorylation of PERK	UMSCC-1, -5PT, -10BPT and -23 cell lines	[98]
Cantharidin (N)	Phosphorylation of eIF2α; Increased CHOP protein levels	human tongue squamous carcinoma-derived SAS, CAL-27 and SCC-4 cell lines.	[99]
Carfilzomib (S)	Phosphorylation of PERK and eIF2α; Increased ATF4 protein levels	The human UMSCC-22A, -22B, -1, 1483 and Cal33 cell lines	[100]
Celastrol (N)	Induced splicing of XBP1; Increased CHOP, GADD34, ERDJ4, HERPUD1 gene expression	The human floor of mouth squamous cell carcinoma lines UMSCC-1, -14A and laryngeal squamous cell carcinoma cell line UMSCC23. The tongue carcinoma cell line CAL27, the salivary epidermoid carcinoma cell line A-253	[101]
Celecoxib (S)	Increased CHOP, GRP78, XBP1 gene expression	SNU-1041 cell line	[102]
Ceramide (N)	ATF6 cleavage; Increased CHOP gene expression; Increased GRP78 protein levels	UMSCC-1 (retromolar trigone/floor of the mouth), UMSCC-14A (SCC of anterior floor of the mouth) and UMSCC-22A (SCC of hypopharynx) cell lines	[103]
Dasatinib (S)	Phosphorylation of eIF2α; Increased CHOP mRNA expression	Ca9-22 (Gingival SSC), SAS, HSC-3 (squamous cell carcinoma of the tongue) cell lines	[104]
DIM (3,3′-diindolylmethane) (N)	Increased CHOP protein levels	SCC9, SCC15 and SCC2095 human OSCC cell lines	[105]
Emodin (N)	Increased GRP78 and CHOP protein levels	Human tongue squamous cancer SCC-4 cell line	[106]
Bortezomib/Romidepsin (S)	Increased CHOP protein levels	EBV-positive nasopharyngeal carcinoma (NPC) cell lines, HA and C666-1,	[107]
Erlotinib (S)	Increased GRP78 protein levels	FaDu cell lines	[90]
Erufosine (S)	Phosphorylation of PERK and eIF2α; Increased ATF6, ATF4 and IRE-1α, CHOP, GRP78, XBP1, ATF3 protein levels; Increased CHOP, ATF3, ATF4 gene expression	HN-5 and SCC-61 cell lines.	[108]
GL63 (=curcumin analog) (S)	Increased CHOP, XBP1 and ATF4 protein levels	Human nasopharyngeal CNE2 cell line	[109]
HMJ-38 (S)	Phosphorylation of eIF2α; Increased GRP78, CHOP and ATF6 protein levels	Human oral carcinoma CAL 27 cell line	[110]
Lobophorins (N)	Increased *XBP1 splicing; Increased CHOP*, *GADD34, ATF3 and GRP78 gene expression*	Human floor of mouth (OSCC) UMSCC-1, -14A cell lines	[111]
Oprozomib (S)	Phosphorylation of PERK and eIF2α; Increased ATF4 protein levels	Human UMSCC-22A, -22B, -1, 1483 and Cal33 cell lines	[100]
PP-22 (N)	Increased PERK, CHOP, GRP78, PDI, ERO1α, IRE1 protein levels	Human nasopharyngeal CNE2 cell line	[112]
Patulin (N)	Increased GRP78, ATF4, CHOP, ATF3 and GADD34 gene expression; Increased XBP1 splicing; Increased CHOP protein levels	Human floor of mouth UMSCC-1, -14A cell lines; Laryngeal UMSCC-23, tongue CAL27 and Pharyngeal FaDu cell lines	[113]
Polydatin (N)	Increased XBP1s, ATF4 and CHOP protein levels	Human nasopharyngeal CNE cell line	[114]
Resveratrol (N)	Increased protein levels of, IRE1, ATF6, CHOP; Phosphorylation of PERK	Human nasopharyngeal NPC-TW076 and NPC-TW039 cell lines	[115]
Retinoid N-(4-hydroxyphenyl) retinamide (S)	Induced XBP1 mRNA splicing; Increased CHOP, GRP78, ATF3, PDIA3 gene expression; Increased CHOP, GRP78 protein levels	Human pharyngeal UMSCC-22A and -22B cell lines	[116]
Rhein (N)	Increased GRP78, CHOP, PERK protein levels	Nasopharyngeal carcinoma-derived cell line NPC-039	[117]
Sulfonamidebenzamide (S)	Increased CHOP, GADD34 gene expression; Increased XBP1 splicing	Human oral UMSCC-23, HN12, HN30 cell lines	[118]
Tetrandrine (N)	Increased protein expression of GADD153, GRP78, ATF-6α	Human nasopharyngeal carcinoma NPC-TW 076 cell line	[119]

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
