# Peer review of "Impact and Relevance of the Unfolded Protein Response in HNSCC"

_ijms, 2019, doi:10.3390/ijms20112654_

Round 1

Reviewer 1 Report

The review by Pluquet et al is a nice summary of the UPR and its significance in cancer research. It suits the interests of IJMS and can be published. However, it requires an extensive formatting. Attached you may find the my comments at few places where corrections are suggested but other than those mentioned, there could be several of such places needing corrections.  

Major Concerns:

There are several reviews available on UPR may be not in particular for HNSCC. I recommend mentioning the existing reviews and highlighting the new points in this review which makes the reader to follow the story with more interest. 

Author Response

Reviewer 1 comments:

The review by Pluquet et al is a nice summary of the UPR and its significance in cancer research. It suits the interests of IJMS and can be published. However, it requires an extensive formatting. Attached you may find the my comments at few places where corrections are suggested but other than those mentioned, there could be several of such places needing corrections. 

Major Concerns:

There are several reviews available on UPR may be not in particular for HNSCC. I recommend mentioning the existing reviews and highlighting the new points in this review which makes the reader to follow the story with more interest. 

We thank the reviewer’s suggestions. We agree that many general reviews on UPR and covering different aspects of cancer have been published. In our manuscript, we mentionned several reviews from different groups :

-Cole DW et al. 2019 Exp Cell Res, in Press

-Song M and Cubillos-Ruiz JR 2019 Trends Immunol 40, 128-141

-Almanza A et al. 2019 FEBS J 286, 241-278

-Hetz C and Papa FR 2018 Mol Cell 69, 169-181

-Wang M et al. Crit Rev Oncol Hematol 2018 127, 66-79

-Papaioannou A and Chevet E 2018 414, 159-192

-Avril T et al. Oncogenesis 2017 6, e373

In particular, Cole DW et al. 2019 Exp Cell Res, is a review very recently published, talking about HNSCC and UPR targeting. This corresponded to the reference 94 in our revised manuscript. We also added others references as requested, including :

In section 1 : ref 5, Cancer Genome Atlas Network, Nature 2015, 517, 576-582

In section 2 : ref 14 Wang WA et al. 2014 Biochim Biophys Acta 1843, 2143-2149

The originality of our review is to update our knowledge about the UPR in HNSCC (specifically). Moreover, we highlight the role of the UPR corresponding to the specific aspects of HNSCC physiology, an entire section (3) is devoted to it, as well as the impact of the UPR in the specific treatments of patients with HNSCC (section 4). Our Table 1 summarizes the Natural compounds/drugs showing activation of the UPR-dependent cell death in HNSCC. We believe that these specific points are of broad interest to the readers interested in UPR and  HNSCC.

Moreover, typos grammar of the manuscript has been thoroughly reviewed and reference style has been fixed

Other concerns :

Page 2, l.66 , the three arms of UPR were introduced and their description is referred to Figure 1.

Reviewer 2 Report

1. The review enumerating the HNSCC and UPR was very well written.

2. All the arms of UPR and their role in this type of cancer are explained.

3. The review also defines the therapeutic angle in this cancer pulling in the UPR which is one of the contemporary anti cancer therapeutics.

4.Overall, I see now flaw and the present review can be straight accepted.            

Author Response

Reviewer 2 comments:

1. The review enumerating the HNSCC and UPR was very well written.

2. All the arms of UPR and their role in this type of cancer are explained.

3. The review also defines the therapeutic angle in this cancer pulling in the UPR which is one of the contemporary anti cancer therapeutics.

4.Overall, I see now flaw and the present review can be straight accepted. 

We thank the reviewer’s comments. Please find enclosed a revised version of our manuscript. Typos grammar of the manuscript has been thoroughly reviewed and reference style has been fixed.